# Parasiticides: Weapons for Controlling Microbial Vector-Borne Diseases in Veterinary Medicine; The Potential of Ethnobotanic/Phytoparasiticides: An Asset to One Health

**DOI:** 10.3390/antibiotics12020341

**Published:** 2023-02-06

**Authors:** Rita Carvalho da Silva, Leonor Meisel, Nóemia Farinha, Orlanda Póvoa, Cristina De Mello-Sampayo

**Affiliations:** 1Regulation and Evaluation of Medicines and Health Products, Faculdade de Farmácia da Universidade de Lisboa, 1649-003 Lisbon, Portugal; 2Laboratory of Systems Integration Pharmacology, Clinical and Regulatory Science, Research Institute for Medicines (iMED.Ulisboa), 1600-277 Lisbon, Portugal; 3VALORIZA—Research Centre for Endogenous Resource Valorization, Polytechnic Institute of Portalegre, 7300-110 Portalegre, Portugal; 4Instituto Politécnico de Portalegre, 7300-110 Portalegre, Portugal; 5Laboratory of Neuroinflammation, Signaling and Neuroregeneration, Research Institute for Medicines (iMed.ULisboa), Faculty of Pharmacy, Universidade de Lisboa, 1649-003 Lisbon, Portugal

**Keywords:** ectoparasites, phytoectoparasiticides, microbial vector diseases, one health

## Abstract

Some ectoparasites are vectors of illness-causing bacteria and viruses, and these are treated with antibiotic and antiviral drugs, which eventually contribute to the excessive use of antimicrobials. Therefore, the control of ectoparasites is crucial, and the challenge will be to manage them in a sustainable way. Data from a preliminary ethnobotanical survey was reanalyzed to obtain information on the use of various plant species in companion animals and livestock as ectoparasiticides. The survey responses were reviewed for traditional use of plants as ectoparasiticides, and cross-sectional bibliographic research was undertaken. The following plants were selected among the nine mentioned plants: *Juglans regia*, *Daphne gnidium* and *Ruta graveolens*, which have the most potential to be developed as veterinary ectoparasiticides. Moreover, the absence of published data for *Plantago lanceolata* and *Cistus populifolius* suggests that their traditional use as ectoparasiticides is noted here for the first time. In summary, these plants could give promising plant-derived veterinary ectoparasiticides that, ultimately, will help reduce and even avoid the excessive use of antimicrobials.

## 1. Introduction

The domestication of animals was part of human history and its evolution since the late Mesolithic and Neolithic periods [1]. This tendency has continued to grow, for companion animals and livestock. Today, with the increase in the world population and the need to satisfy a diverse array of goods, for instance, meat, milk, and eggs, the importance of livestock has grown, characterised by their abundance, raised production systems, and distribution of commodities. In the meantime, the number of companion animals increased, alongside urbanization, i.e., there are more and more companion animals in family homes, where they cohabit with several people in the same house and also in public places. This increased proximity of relations has brought some negative aspects [1,2]. Although both types of animals come from different habitat types, one prevalent negative point emerging from these human/animal interactions is the ectoparasite infestation and, consequently, the enhancement of human parasitism [3,4,5]. This means that one parasite gains coexistence with the host, whom is negatively affected [6].

Ectoparasites or external parasites consist of multicellular beings found under or on the skin of the host [6,7,8]. All animals have contact with ectoparasites at some point in their lives. Depending on the type of association of the ectoparasite with the host, i.e., if it needs the host to survive and reproduce or if it uses the host only to feed, it may have different impacts on the life of the animal [9]. The main problem caused by ectoparasites in companion animals is discomfort, which increases according to the number of parasites present [10,11]. However, the consequences are far more significant and worrying for livestock. When infected, this can lead to substantial losses due to weight loss, reduced milk production or skin damage, with high economic impacts, since it is often not possible to use the animal as a food source [12,13]. In both cases, there is also an added main concern with the presence of ectoparasites in animals and their role as a vector of several pathogens (viruses, bacteria, helminths, or protozoa) [14].

The proper treatment and control of ectoparasites is a recognized veterinary target with medical repercussions for animal and human health [10,14,15]. Therefore, it is essential to have an approach that integrates people, animals, plants, and their shared environment, using the so-called one health approach [16]. Furthermore, by reducing the number of ectoparasites present in the lives of animals, the number of vector-borne diseases (VBDs) and the consequent use of antimicrobials will be reduced and eventually even avoided, given that these diseases are usually treated with antibiotics. Table 1 shows the association between ectoparasites (vector) and their transmitted pathogens, which originate from the VBD. Research on this issue and statements from the European Scientific Counsel Companion Animal (ESCCAP) achieved that fleas, mosquitoes, ticks, and mites are the ectoparasites with the most significant impact on the transmission of VBD and zoonotic diseases [17,18].

Medicines (called antiparasitics) have been developed for companion and livestock animals to prevent and treat ectoparasitic diseases. The active substances are similar for both groups; however, the posology features regarding the animal species and group have been shielded. According to Regulation (EU) 2019/6 of the European Parliament and of the Council of 11 December 2018 on veterinary medicinal products and the repealing Directive 2001/82/EC, an antiparasitic was defined as: “a substance that kills or interrupts the development of parasites, used to treat or prevent an infection, infestation or disease caused or transmitted by parasites, including substances with a repelling activity” [31].

Most parasiticides used to control ectoparasites are of synthetic origin, and some of them could still be used as pesticides in agricultural areas [3]. Thus, these substances have raised concerns about environmental impact, human health (e.g., food residues), and the possible emergence of resistance [3,32]. Considering the necessity to control ectoparasites, it is essential to act to address the concerns raised by chemically synthesized ectoparasiticides. One option is to review the knowledge about plants, and their derivatives, passed down over the centuries. The development of plant-based therapies is a solid alternative for standard, agroecological, and holistic farming systems; it is also an important ally for the combat of drug-resistant parasite populations [33]. Phytochemicals from medicinal plants used for the cure and prevention of diseases like scabies and myiasis are safe, cheap, and less susceptible to resistance. Their use can facilitate the discovery of newer drug molecules for the eradication of other ectoparasites such as lice, ticks, mites, etc., and alleviate the diseases caused by them [34].

Historically, plants have accompanied human evolution, and have been used in different areas, such as treating and preventing diseases in people and animals [13]. For example, the use of plants to control ectoparasites has been present in populations for centuries [35]. However, this information is not documented; it is passed on verbally from generation to generation, thus running the risk of being lost [36]. It is essential to highlight that those plant-derived products (PDPs) are a strong candidate to counter, for instance, environmental problems caused by misuse/abuse of synthetic ectoparasiticides, and a sustainable alternative to control ectoparasites, the diseases they transmit and, indirectly, the use of antimicrobials. Expanding the research in this area is currently a challenge.

Given that Portugal is a country rich in plant biodiversity, ethnobotany and ethnopharmacology represent crucial instruments for the bioprospection of its traditional valorization. Furthermore, PDPs in veterinary treatments have increased in importance, since plants, particularly endemic ones, have proven effective and less harmful than chemical products [12].

Based on a previous ethnobotanical survey, a new analysis arrangement of its interviews (survey form Table A1 in Appendix A) was performed to obtain information on the use of various plant species in dogs and livestock. Although during the bibliographical research, it was noted that there are diseases that may be associated with other animals besides those addressed here, these were not cited since there was no data from the surveys. Nevertheless, their importance and impact on the animals’ and people’s lives are also significant. For example, *Dermanyssu gallinae* was detected in 95.8% of farms in Portugal [37]. *Dermanyssus gallinae* or *Ornithonyssus sylviarum* infect laying hens, while *D. gallinae* is increasingly implicated in human infestation and as being competent vectors of disease [37]. From this framework, the ovine psoroptic mange (sheep scab) should be highlighted, which does not affect humans being transmitted via sheep-to-sheep or sheep–environment contact. Nowadays, it represents a cost-effective issue for farmers [38].

The immediate main objective of this paper is to select plant species based on their ethnopharmacology to reach alternative and sustainable phytomedicines to be used as ectoparasiticides. Indirectly, this also avoids the further use of antimicrobials to treat VBDs by the control of ectoparasites.

## 2. Results and Discussion

### 2.1. Data Considering Plant Phytoectoparasiticide Potential and Animal Species Mentioned

The reanalysis of the ethnobotanical survey data demonstrated that, among the 933 citations, considering the reference to a plant–animal—treatment, only 3.7% stated the benefit of plants in treating ectoparasitism [39]. Moreover, the citations of plant use for ectoparasitism treatment by different animal species were indicated for *Canidae* with 27%, followed by *Equidae* (23%), while *Suidae* and *Capridae* showed 15%, as depicted in Figure 1.

### 2.2. Plant Species Selected with the Most Ectoparasiticide Potential

#### 2.2.1. Plant Species Mentioned in the Ethnobotanical Survey

Among the 18 botanical species quoted in the traditional use survey with veterinary medicinal applicability, where *Malva* sp. and *Lavatera* sp. were the most cited, nine plants were mentioned for their ectoparasiticide potential for dogs and livestock [39]. Table 2 lists these plants in descending order of the number of times mentioned in the various interviews.

This data was obtained through interviews in which people over 70 participated, proving that plants have been used for several years to control ectoparasites. The knowledge of using them efficiently has remained in some populations; however, there is a strong tendency for this knowledge to be lost due to easy access to other conventional medicinal drugs [40]. Since this kind of knowledge allows the valorization of native and endemic species of the regions, it has an added value to preserve it.

#### 2.2.2. Refined Plants Selection through Bibliographic Research

The results obtained from the countryside population’s knowledge regarding ethnoectoparasiticide use led us to carry out, through bibliographical research, the evaluation of each plant’s efficacy related to its characteristics as an ectoparasiticide. The selected plants and their main ectoparasiticide potential characteristics are described in Table 3.

The bibliographic research enabled us to merge and deepen our knowledge of the selected plant species mentioned in the survey as ectoparasiticides. Nevertheless, for the species *Plantago lanceolata* and *Cistus populifolius*, there was no information indicating their use in the control of ectoparasites, nor on their phytochemistry characterization. As far as we know, it is the first time these two plants have been indicated as having ectoparasiticide potential, as has been highlighted by the survey respondents. The species *Cistus ladanifer* only presents traditional information on its use for controlling flies, but provides no information on its phytochemistry characterization, which is not sufficient [62].

*Juglans regia*, commonly known as the walnut tree, is a plant with great potential in various therapeutic areas [70]. It is a plant with a broad worldwide distribution. It is found in Asia (the foothills of the Himalayas, Iran, China and Japan), Southern and Eastern Europe, and North and South America [70]. Different parts are used according to the treated affection. The Entomological Society of America published a paper showing that leaf extracts are toxic against mites through physical contact, as well as through the systemic route [42]; the compounds present are terpenes, hydrocarbons, and esters, as well as strong antioxidant components (flavonoids and phenolic compounds) [39,42,44,70]. There are studies where lethality has been confirmed, under laboratory conditions, for mites (*Tetranychus cinnabarinus* and *Tetranychus viennensis*). According to the ethnobotanical perspective, it has been indicated as a mosquito repellent [44]. The possibility of lethal outcomes against adult ticks was mentioned in the present study’s survey. Furthermore, laboratory studies confirm the impact on their larval stage [39]. In 2021, the efficacy of extracts prepared with the green hull of *Juglans regia* was tested against the acaricidal activity, more specifically, *Rhipicephalus microplus larvae* [44]. It was observed that the extracts have significant larvicidal activity, since, depending on the concentration of the different extracts evaluated, a 100% mortality rate was achieved [44].

Although *Mentha pulegium* has been mentioned as being used in mixtures with *Olea europea*, there is research on its individual use too. It is present in temperate and subtropical regions where it has been cultivated and naturalized [47]. Some studies point out that extracted essential oils have a mighty mosquito larvicidal potential and activity against mites and flies [47,48,49]. Pulegone is the main component responsible for mosquito larvicidal activity [47].

*Olea europaea*, commonly known as the olive tree, is distributed throughout the five continents, being more predominant in regions with a Mediterranean climate [71]. Based on traditional olive tree use, some studies have shown that it effectively repels mosquitoes when burned directly in the early evening [51]. However, its repellent effect on mosquitoes and house flies needs further study besides the direct burning of vegetative materials [52,62]. The compounds in leaves are phenolic compounds, fatty acids, terpenoids, alcohols and sterols, hydrocarbons, and carbohydrates. The phenolic compounds may be responsible for the antiparasitic activity [42,52].

*Daphne gnidium*, commonly known as flax-leaved daphne, is only found in some European countries, such as Portugal, Albania, the Balearic Islands, Corsica, France, Greece, Spain, Gibraltar, Italy, Sardinia, Sicily, the Canary Islands, Morocco, Algeria, Tunisia, and Turkey [72]. *Daphne gnidium* has been cited for external use in animals as an effective repellent against lice, fleas, ticks, leeches, and insects [54]. In addition, its roots and branches have been reported to be used to control flea infestations at home [73]. Furthermore, several secondary metabolites have been identified in essential oils and extracts, including terpenoids, coumarins, flavonoids, fatty acids, and alkanes [55]. However, it is still unclear which chemical compounds in the plant are responsible for the ectoparasiticide effect [54]. Although the current information comes from traditional knowledge-gathering, some ethnobotanical studies mention this plant as being effective against fleas. As the flea is a key parasite to be controlled, it is essential to analyze and corroborate the information on *Daphne gnidium.*

*Nicotiana tabacum*, commonly known as tobacco, has a very high global prevalence on all five continents, quite pre-eminently. Over the years, tobacco has been cited several times as an effective pesticide and used as an insect or tick repellent, thus preventing vector-borne diseases [45,74]. In recent years some studies have been conducted to verify the effectiveness against the tick *Rhipicephalus* sp. (more specifically, *R. microplus*, *R. sanguineus*). The larvicidal and adulticidal activity of *N. tabacum* leaf extracts have been proven [45,58]. This plant presents bio-compounds that have been proven effective against some parasites (fleas, fly larvae and ticks). Still, its primary alkaloid, nicotine, is detrimental to the environment [58]. In addition, it has been used excessively without awareness of its toxicity profile, impacting pollinators, aquatic communities, and soils [58,73].

*Ruta graveolens*, commonly known as rue, has a worldwide distribution. However, its prevalence is more significant in the European continent and some areas of the American continent [75]. In addition, some studies have confirmed lethality under laboratory conditions in adult ticks and fleas [65,66,67,68]. Therefore, in compliance with the ethnobotany, it has been indicated against several species of adult mosquitoes and containing larvicidal activity [68,69]. The main components of extracts obtained by maceration are alkaloids, coumarins and saponins [68]. Therefore, in compliance with the present ethnobotany survey, it has been indicated against several species of adult mosquitoes and containing larvicidal activity [68,69]. One of the benefits of choosing the *Ruta graveolens* species is the proven activity of leaf extracts as an adulticide of *Rhipicephalus microplus*, so it is likely to have efficacy against other tick species [64,65,66,67]. In addition, efficacy against *Ctenocephalides canis* has been described [64]. However, some studies show that this plant species may be toxic to humans. Therefore, it is essential to evaluate extraction methods, isolate compounds to be used, and verify the possibility of dilution before application [65,76].

Concerning the plant species underlined as ectoparasiticides reported in the present ethnobotany survey and the bibliographic research, *Juglans regia*, *Ruta graveolens* and *Daphne gnidium* have the most significant potential to be developed as ectoparasiticidal veterinary medicines. Table 4 displays the ectoparasite potential of the elected plants deserving future veterinary ectoparasiticide development.

An ectoparasiticide should ideally contain a broad spectrum of activity comprising adulticide, larvicide and repellent properties affecting many ectoparasite species. However, the larvicidal characteristic is significant because it interferes at the beginning of the parasite cycle, inhibiting the full development of the parasite. Moreover, a practical effect on ticks is also essential regarding their pathogenic power and associated diseases and VBDs.

Therefore, the selection of these three plants is based mainly on their broad spectrum of activity (e.g., larvicide) and the wide range of ectoparasites they affect. Additionally, *Junglas regia* and *Ruta graveolens*, with a wide distribution worldwide, offer excellent production and harvesting availability, thus causing less ecological impact when harvesting the plant to produce the ectoparasiticides [42]. Nevertheless, there is evidence that the chemical composition of these plants may differ according to the climate and soil. For this reason, it is crucial to have a comprehensive knowledge of the selected plant species’ chemotypes to ensure a lower variability in chemical profiles. Hence, the final product’s bioactivity is predictable. Since there are reported cases where geographical location, seasonality and harvest year can influence the final product’s effectiveness, selecting strains that are less likely to have composition variability, and hence alter the final product, is essential. In addition, storage can cause changes in product properties, so it is crucial to ensure good and adequate storage conditions [76]. In some cases, toxicity has been reported, although most PDPs do not exhibit toxicity. However, it is necessary to consider the amount used, the method of extraction, isolation of the compound required, possible dilution of the compound and the specimen of the species chosen [76]. Even though *Daphne gnidium* is not as promising as the other two selected plants, its growth being confined to some European countries makes it a potential plant to be further valorized. In addition, at the traditional use level, it is cited as an excellent way to control fleas, an ectoparasite that is a vector for many diseases. The challenge will be to control, with a sustainable approach, ectoparasites for vector-borne diseases, and therefore controlling those VBDs to avoid antimicrobial use.

## 3. Methods

### 3.1. Data Source

The data source was a preliminary ethnoveterinary survey based on 56 semi-structured interviews performed in the Alentejo area, Portugal, between July 2011 and April 2012. The profile of respondents was male, over 70 years, with a primary school level of education and professions related to agriculture. This previously conducted inquiry intended to find relevant plants that could be searched ahead for introduction into animal, or even in pastures, for medicinal purposes [39,76]. However, in the present study, this data was revisited with a different perspective.

### 3.2. Data Analysis and Data Selection

The data from the preliminary ethnobotany survey were reanalyzed, envisioning the objective of the present manuscript, which is to obtain information on the use of various plant species in companion animals and livestock as ectoparasiticides. Thus, a new arrangement of the survey interviews’ responses was performed, considering the following features: (1) plant species that were mentioned as being used to control ectoparasites; and (2) the specific use of plants to control ectoparasites in *Canidae*, *Bovinae*, *Caprinae*—*Capra* and *Caprinae*—*Ovis*. Figure 2 illustrates the present schematic representation of the collected responses.

### 3.3. Bibliographic Research

A cross-sectional bibliographic review was undertaken using scientific search engines such as Google Scholar, ScienceDirect, SpringerLink, Elsevier and PubMed between 2020 and 2022. The main objective of this approach was to identify the state of the art concerning the ethnobotany/phytoectoparasiticides in the veterinary area. Thus, a combination of individual search terms and strings was used:

“*Canidae*”, “*Bovinae*, *Caprinae—Capra* and *Caprinae—Ovis*”, “ectoparasites and vector-borne diseases associated with fleas/ticks/mosquitoes/mites”, “ethnobotanical”, “phytoectoparasiticides and plants designated in the interviews (*Juglans regia*/*Mentha pulegium*/*Olea europaea*/*Daphne gnidium*/*Nicotiana tabacum*/*Cistus ladanifer*/*Plantago lanceolata*/*Ruta graveolens*/*Cistus populifolius)*”, “effectiveness (*Ruta graveolens*/*Daphne gnidium*/*Juglans regia)*”, “efficacy (*Ruta graveolens*/*Daphne gnidium*/*Juglans regia*)”. The Articles/Results were screened by their relevance, considering: (a) the significance of controlling the main ectoparasites of *Canidae* and livestock, which are vectors of disease-causing bacteria and viruses; (b) the relevant information regarding the use of ectoparasiticide properties of plant species; and (c) selection of plants with the highest efficacy against ectoparasites and, therefore, the highest probability of effectiveness.

The exclusion criteria comprised the articles that did not include the plant species designated in the interviews. The consort diagram of the literature research can be seen in Figure 3.

## 4. Conclusions

Some ectoparasites are vectors of illness-causing bacteria and viruses, and these are treated with antibiotics and antiviral drugs. However, since they have been identified as being responsible for spreading diseases, ectoparasites significantly impact animal health, causing relevant economic losses, especially in domestic animals and livestock, but also compromising human health, and thus eventually contributing to the excessive use of antibiotics and antiviral drugs. Therefore, the control of ectoparasites is crucial.

An alternative, which are being increasingly tested, are the compounds derived from plants that mostly are identified based on the traditional knowledge of different populations. This alternative has advantages and disadvantages; on the one hand, it can solve some of the problems of ectoparasites currently used; however, on the other hand, few studies validate its actual effectiveness.

By reanalyzing the data from an ethnobotanical survey study carried out in the Alentejo region, Portugal, regarding the traditional use of various plant species in the treatment of animals and searching databases of studies publications with those plants, it was possible to select three out of the nine plants reported as having ectoparasiticide potential. The species *Juglans regia*, *Daphne gnidium* and *Ruta graveolens* seem to have potential ectoparasiticidal efficacy with reduced environmental impact and low risk for humans and animals. That said, toxicity, efficacy, stability, and safety studies should be conducted to prove the possibility of using extracts from these different plant species to develop antiparasitic medicinal products. Moreover, as far as we know, the absence of published data for the two species *Plantago lanceolata* and *Cistus populifolius*, suggests that their traditional use as parasiticide is noted here for the first time.

## Figures and Tables

**Figure 1 antibiotics-12-00341-f001:**
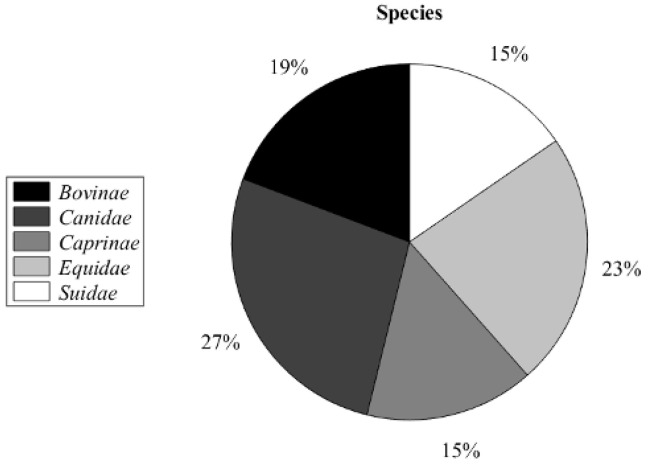
Percentage of phytoectoparasiticides use by different animal species.

**Figure 2 antibiotics-12-00341-f002:**
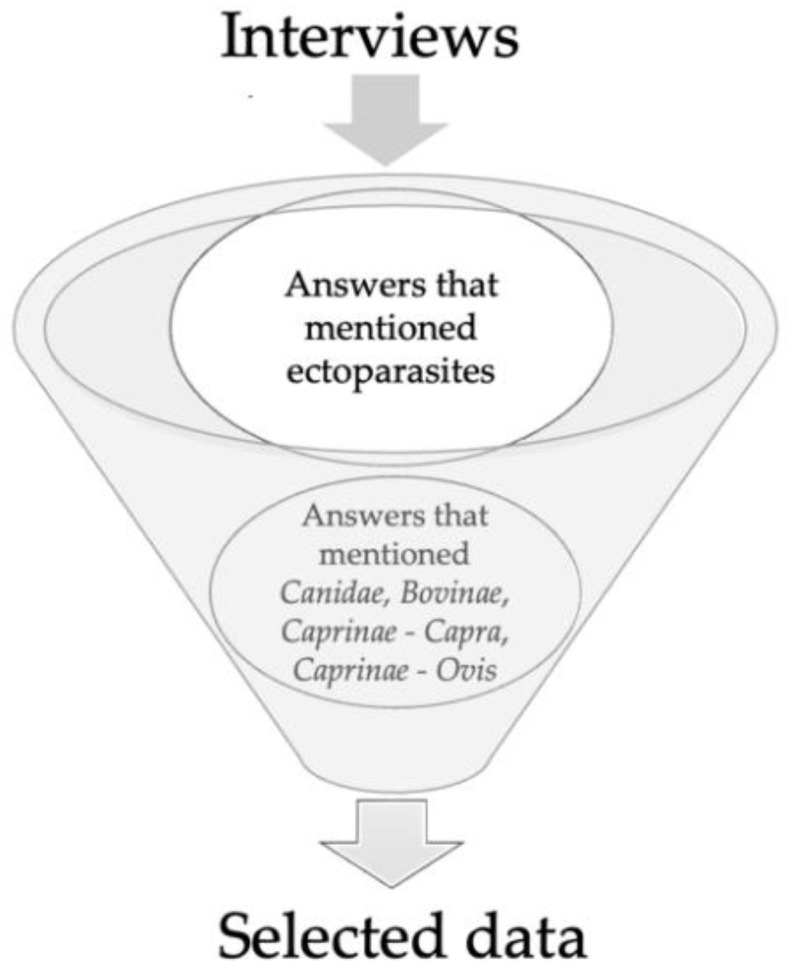
Schematic representation of data selection from the collected survey responses.

**Figure 3 antibiotics-12-00341-f003:**
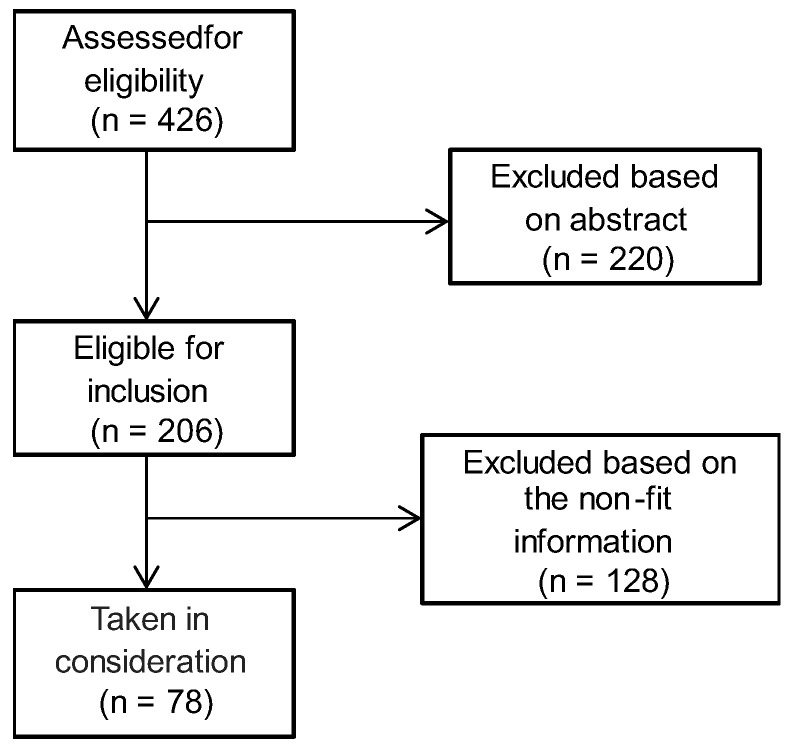
Consort diagram of the literature research.

**Table 1 antibiotics-12-00341-t001:** Viruses and bacteria-associated diseases or infections caused by VBDs.

Arthropod	Vector	Type of Pathogen	Pathogens Transmitted	Associated Disease/Infection	References
Ticks	*Dermacentor variabilis*, *D. andersoni*, *A. cajennense*, *Rhipicephalus sanguineus*, others	Bacteria	*Rickettsia rickettsii*	Rocky Mountain spotted fever	[14,15,19,20]
	*Amblyomma maculatum* group		*Rickettsia parkeri rickettsiosis*	Rickettsia parkeri	
	*Amblyomma americanum*		*Ehrlichia chaffeensis*	Human ehrlichiosis	
	*Amblyomma americanum*		*Ehrlichia ewingii*	Human ehrlichiosis	
	*Ixodes scapularis* and *Amblyomma americanum*		*Ehrlichia muris eauclairensis*	Ehrlichiosis	
	*Ixodes ricinus*, *I. trianguliceps*		*Anaplasma phagocytophilum*	Anaplasmosis (granulocytic ehrlichiosis)	
	*Ixodes persulcatus*		*Anaplasma capra*	Human anaplasmosis	
	*Rhipicephalus* spp., *Dermacentor* spp.		*Anaplasma ovis*	Human anaplasmosis	
	*Ixodes ricinus*, *I. persucatus*		*Neoehrlichia mikurensis*	Neoehrlichiosis	
	*Ixodes* spp.		*Coxiella burnetii*	Coxiellosis (Q fever)	
	*Ixodes scapularis*, *I. ricinus*, *I. pacificus*, *I. persulcatus*, others		*Borrelia burgdorferi*, *B. afzelii*, *B. garinii*, *B. bissettii*	Lyme disease	
	*Ornithodoros* spp.		*Borrelia* spp.	Tick-borne relapsing fever	
	*Ixodes ricinus*, *I. scapularis*, *I. pacificus*, *I. persulcatus*		*Anaplasma phagocytophilum*	Tick-borne fever	
	*Haemaphysalis leporispalustris*, others		*Francisella tularensis*	Tularemia	
	*Amblyomma* spp.		*Rickettsia africae*	African tick bite fever	
Ticks	*Ixodes ricinus*, *I. trianguliceps*, *I. persulcatus*	Virus	TBE virus	Tickborne encephalitis	[10,21]
	*Hyalomma* spp.		CCHF virus, Bunyaviridae, Nairovirus	Crimean—Congo hemorrhagic fever	
Fleas	*Ctenocephalides felis*	Bacteria	*Bartonella henselae*	Bartonellosis (Cat scratch disease)	[1,15,22,23]
	Several fleas		*Coxiella burnetii*	Q fever	
	Several fleas		*Francisella tularensis*	Tularemia	
	*Xenopsylla cheopis*, *Ctenocephalides*		*Rickettsia typhi*	Murine typhus	
	*Ctenocephalides felis*		*Rickettsia felis*	Rickettsiosis	
	*Mainly Xenopsylla*		*Yersinia pestis*	Plague	
Mites	*Liponyssoides sanguineus*	Bacteria	*Rickettsia akari*	Rickettsialpox	[10,24,25,26]
	*Ornithonyssus bacoti*		*Coxiella bumetti*	Q fever	
	Chigger mite—*Trombicula akamushi*		*Orientia tsutsugamushi*	Scrub typhus	
Mosquitoes	Mosquitoes—*Aedes* spp.	Virus	Rift Valley virus, *Bunyaviridae*, Phlebovirus	Rift Valley fever	[7,10,17,27,28,29,30]
	*Culex* spp. and other mosquitoes		West Nile virus (WNV)	West Nile virus infection	
	*Culex* spp., *Aedes* spp. and *Anopheles* spp.	Bacteria	*Francisella tularensis*	Tularaemia	[17,29,30]

**Table 2 antibiotics-12-00341-t002:** Plants mentioned in the ethnobotanical survey with ectoparasiticide potential for dogs and livestock.

Species	Family	Animal Species	Number of Mentions
*Juglans regia* L.	*Juglandaceae*	*Canidae*, *bovinae*, *caprinae—capra*, *caprinae—ovis*, *equidae*, *suidae*	6
*Mentha pulegium* L.	*Lamiaceae*	*Bovinae*, *caprinae—ovis*, *suidae*	3
*Olea europaea* L.	*Oleaceae*	*Bovinae*, *caprinae—ovis*, *suidae*	3
*Daphne gnidium* L.	*Thymelaceae*	*Bovinae*, *equidae*	2
*Nicotiana tabacum* L.	*Solanaceae*	*Canidae*	2
*Cistus ladanifer* L.	*Cistaceae*	*Canidae*	1
*Plantago lanceolata* L.	*Plantaginaceae*	*Canidae*	1
*Ruta graveolens* L.	*Rutaceae*	*Canidae*	1
*Cistus populifolius* L.	*Cistaceae*	*Canidae*	1

**Table 3 antibiotics-12-00341-t003:** Plants species with ectoparasiticide potential for dogs and some livestock.

Plant Species	Common Name	Geographic Distribution	Targeted Vector	Extracted Compounds	Activity
*Juglans regia*	Walnut tree [41]	Worldwide [41]	Mite (*T. cinnabarinus* and *T. viennensisin*), Mosquito and *Rhipicephalus microplus* [42,43,44]	Terpenes, hydrocarbons, esters, and strong antioxidant components such as flavonoids and phenolic compounds [42,44,45]	Acaricide and repellent [42,43,44]
*Mentha pulegium* (with *Olea europaea*) ^1^	Pennyroyal [46]	Native to almost all of Europe. Appears in other parts of the world [46]	Mosquito larvae, house fly, and mite [47,48,49]	Pulegone, piperitone, menthol, menthone and piperitone oxide [47]	Larvicide, adulticide, acaricide and repellent [47,48,49]
*Olea europaea* (with *Mentha pulegium*) ^1^	Olive tree [50]	Worldwide. More predominant in regions with a Mediterranean climate [50]	Mosquito [51]	Phenolic compounds, fatty acids, terpenoids, alcohols and sterols, hydrocarbons, and carbohydrates [42,52]	Repellent [51]
*Daphne gnidium*	Flax-leaved daphne [53]	South Europe, North Africa and Canary Islands [53]	Louse, flea, tick, and insect [54]	Terpenoids, coumarins, flavonoids, fatty acids, and alkanes [55]	Acaricide, adulticide and repellent [54]
*Nicotiana tabacum*	Tabak, Tabaco, Tobacco [56]	Worldwide [57]	Tick larvae (*Rhipicephalus sanguineus* and *Rhipicephalus* sp.), tick adult (*Rhipicephalus* sp.), fly larvae (*Musca domestica*), and flea [45,58,59,60]	Alkaloids—nicotine, saponins, tannins, flavonoids, terpenoids, anthraquinones and steroids [53,61,62]	Acaricide, larvicide, adulticide and repellent [45,58,59,60]
*Cistus ladanifer*	Rockrose [61]	Occidental Mediterranean region [61]	Fly [62]	No relevant data was found to prove its use	No relevant data was found to demonstrate its use
*Plantago lanceolata*	Narrow-leaf plantain, Ribwort plantain [63]	Worldwide [63]	No relevant data was found to prove its use	No relevant data was found to prove its use	No relevant data was found to demonstrate its use
*Ruta graveolens*	Common rue [64]	Worldwide [64]	Tick adult, flea, adult and larvae mosquito [65,66,67,68,69]	Alkaloids, coumarins and saponins [68]	Acaricide, adulticide and larvicide [65,68]
*Cistus populifolius*	Poplar-leaved rock rose, Rockrose [70]	Iberian Peninsula, South France and North Marroco [70]	No relevant data was found to prove its use	No relevant data was found to demonstrate its use	No relevant data was found to demonstrate its use

^1^ Quoted as used in mixtures.

**Table 4 antibiotics-12-00341-t004:** Overview of selected plants with ectoparasiticide potential.

Plant Specie	Ectoparasite
*Juglans regia*	Adult and larvae Ticks Mites Mosquitoes/Flies ^1^
*Daphne gnidium*	Ticks ^1^ Fleas ^1^ Mosquitoes/Flies ^1^
*Ruta graveolens*	Adult Ticks Fleas Adult Mosquitoes/Flies ^1^ Larvae Mosquitoes/Flies ^1^

^1^ Data from traditional knowledge only.

## Data Availability

Not applicable.

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
