# Peer review of "Parasiticides: Weapons for Controlling Microbial Vector-Borne Diseases in Veterinary Medicine; The Potential of Ethnobotanic/Phytoparasiticides: An Asset to One Health"

_antibiotics, 2023, doi:10.3390/antibiotics12020341_

Round 1
Reviewer 1 Report
Thank you for submitting this paper to Antibiotics.
The paper is nicely written and summarises a survey data set, using terms gained from this to form a literature search.
However, there are some omissions from the review which I feel compelled to point out. These omissions may be because you lacked data from your survey which was reanalysed retrospectively but I feel that they should at least be included in the introduction to give a more rounded discussion of the subject of ectoparasites and their control using plant extracts. I feel that your discussion is a bit narrow and too confined to the species flagged up in the survey data-there could be useful information in comparative research.
In the introduction, you state that there is an increase in human parastism but only use one referenced source for this-more evidence is required for this claim (Line 43).
Although you mention eggs, no mention is given to Dermanyssus gallinae or Ornithonyssus sylviarum which infest laying hens and of which, D gallinae in particular are increasingly indicated in human infestation and in being competent vectors of disease. D gallinae is much more important in this regard than Ornithonyssus bacoti, which you have included. D gallinae is also a problem in Portugal (DOI: 10.1080/03079457.2019.1606415).
These mites and other parasites are major welfare issues in farmed animals also and no discussion of this was mentioned and while not the main focus of your study, which was from a One Health point of view, is important as it drives research into potential controls. Even if data on poultry was not available from your data set and this is the reason for the omission, a summary sentence in the introduction and discussion could lend weight to your argument pro-phytoparasiticides and the reason why it was omitted could be stated. A lot of work on phytobotanicals and poultry mites has been published and also sheep scab mite. Sheep scab doesn't infest humans nor is it a One Health concern other than the use of antibiotics to treat secondary infections but the research in the use of plant extracts might assist your discussion.
I think your analyses and discussion after that is fine although there is a lack of discussion on the potential toxicity of these compounds (walnut for example) and what can be done to mitigate that ie have these or similar compounds been successfully detoxified for similar or other uses.
Author Response
We are very grateful for your constructive comments and suggestions concerning our manuscript entitled” Parasiticides: weapons for controlling microbial vector-borne diseases in veterinary medicine. The potential of ethnobotanic/phytoparasiticides: an asset to One Health", ID: antibiotics- 2195663.
our comments are all valuable and very helpful for revising and improving our paper. We have carefully studied the comments and made corrections accordingly, which we hope will meet with approval. You can find our responses to your comments in the attachment.
Please see the attachment. Thank you.

Reviewer 2 Report
The topic treated in the submitted article is of great relevance in the field and could be a useful starting point for future researches investigating the efficacy of plant extracts as ectoparasiticides. The main novelty here can be considered the One Health approach that the researchers kept throughout the entire paper since this is a concept of evergrowing importance nowadays in veterinary, medical and biological science. Tables are well written and useful to give to the reader comprehensive infos (especially table 1) on the treated topics.
Using retrospective survey data there's little to no room for changing methodological aspects of the present work, maybe a supplementary file showing the entire survey form could be added to the paper.
Line 84-85: the sentence is not clear at all, you should rephrase it.
In paragraph 2.3. a consort diagram of the literature research with the specific number of gathered, excluded and finally taken in consideration articles should be added.
Line 151: delete "by"
Author Response
We are very grateful for your constructive comments and suggestions concerning our manuscript entitled” Parasiticides: weapons for controlling microbial vector-borne diseases in veterinary medicine. The potential of ethnobotanic/phytoparasiticides: an asset to One Health". (ID: antibiotics- 2195663).
Your comments are all valuable and very helpful for revising and improving our paper. We have carefully studied the comments and made corrections accordingly, which we hope will meet with approval. You can find our responses to your comments in the attachment.
Please see the attachment. Thank you.
